# Diagnostic Challenges on the Laboratory Detection of Lupus Anticoagulant

**DOI:** 10.3390/biomedicines9070844

**Published:** 2021-07-20

**Authors:** Armando Tripodi

**Affiliations:** 1IRCCS Ca’ Granda Ospedale Maggiore Foundation, Angelo Bianchi Bonomi Hemophilia and Thrombosis Center, Via Pace 9, 20122 Milano, Italy; armando.tripodi@unimi.it; Tel.: +39-02-55035437; Fax: +39-02-54100125; 2Fondazione Luigi Villa, 20122 Milano, Italy

**Keywords:** antiphospholipid antibody syndrome, thrombosis, anticoagulation, activated partial thromboplastin time, dilute Russell viper venom test

## Abstract

Lupus anticoagulant (LA) is one of the three laboratory parameters (the others being antibodies to either cardiolipin or β2-glycoprotein I) which defines the rare but potentially devastating condition known as antiphospholipid syndrome (APS). Testing for LA is a challenging task for the clinical laboratory because specific tests for its detection are not available. However, proper LA detection is paramount for patients’ management, as its persistent positivity in the presence of (previous or current) thrombotic events, candidate for long term anticoagulation. Guidelines for LA detection have been established and updated over the last two decades. Implementation of these guidelines across laboratories and participation to external quality assessment schemes are required to help standardize the diagnostic procedures and help clinicians for appropriate management of APS. This article aims to review the current state of the art and the challenges that clinical laboratories incur in the detection of LA.

## 1. Introduction

Lupus anticoagulant (LA) is part of a heterogenous autoantibody family targeting negatively charged phospholipids (PL) in complex with such proteins as prothrombin, β2-glycoprotein-I (β2-GP-I), and others [1]. The term LA is actually a misnomer, as the presence of LA in plasma results in the prolongation of the clotting times measured by such PL-dependent coagulation tests as the activated partial thromboplastin time (aPTT) and/or dilute Russell’s viper venom test (dRVVT). In contrast, the presence of LA is paradoxically associated with thrombosis (either venous, arterial or both) and/or pregnancy complications [1]. LA is one of the three laboratory criteria that characterize the antiphospholipid syndrome (APS), a relatively rare but potentially devastating condition. The other two criteria defining APS are the presence of antibodies to cardiolipin (aCL) or β2-Glycoprotein-I (aβ2-GP-I). According to the guidelines issued by the International Society on Thrombosis and Haemostasis (ISTH), APS is defined whenever there is the coexistence within the same patient of at least one clinical event (i.e., venous/arterial thrombosis or pregnancy complications) and positivity for LA or the presence of medium-high titers of either aCL or aβ2-GP-I [2]. Triple positivity (i.e., concomitant positivity for LA, aCL and aβ2-GP-I) identifies patients at high risk of clinical events [3]. Patients with APS and persistently confirmed laboratory diagnoses are candidates for long term anticoagulation to prevent recurrent thrombotic events [4]. On the other hand, anticoagulation with any of the drugs currently used [i.e., heparins, vitamin K antagonists (VKA), or direct oral anticoagulants (DOAC)] are burdened with the risk of hemorrhage, which is unacceptable if patients are falsely positive for the laboratory diagnosis of APS. The search for aCL and aβ2-GP-I is relatively simple and reliable, especially when using the last generation of chemiluminescent assays [5]. Details related to their importance for the APS diagnosis, their practice and limitations have been discussed elsewhere [6] and are outside the scope of this review. In contrast, LA detection is burdened with poor standardization, difficult interpretation of results and interference by drugs, which are often prescribed to the investigated patients and that may make laboratory diagnosis difficult. This article aims to review the current laboratory procedures used to detect LA and to discuss merits and drawbacks that may help clinicians and laboratory operators to manage patients with APS.

## 2. Phospholipid (PL)-Dependent Tests to Detect LA

Historically, LA has been detected by the time-honored coagulation tests aPTT, when this test was (accidentally) found to be prolonged on pre-surgical screen in subjects, for whom no deficiencies of coagulation factors were observed and who were otherwise healthy. The earlier findings of aPTT prolongation in the absence of coagulation factor deficiencies was considered as a nuisance because it called for (uneventful) investigation of prolonged aPTT in otherwise (apparently) healthy individuals. This situation prompted the quest for aPTT reagents insensitive to LA. However, it was soon realized that some patients positive for LA had an increased risk of thrombosis and/or pregnancy complications and this attracted the attention of clinicians and laboratory operators. Unfortunately, there were (and there are still) no specific tests to detect LA and therefore the laboratory diagnosis was (and still is) based on PL-dependent coagulation tests combined with diagnostic criteria (see Figure 1).

The diagnostic criteria require three iterative procedures called screen, mix, and confirm. Typically, patients having LA display PL-dependent coagulation tests (see below) longer than the upper limit of the reference range (screen criterion), which are not normalized upon mixing patient plasma with an equal portion of normal plasma (mix criterion) and are shortened by repeating the test upon increasing the PL concentration (confirm criterion). The rational of the screen rests on the fact that LA targets negatively charged PL, which are part of the composition of coagulation tests, making their results to be prolonged. On the other hand, if the prolongation of the screen procedure is due to one (or more) coagulation factor deficiency, the addition of a normal plasma will provide sufficient amounts of the missing factor(s) to the patient plasma and the results of the coagulation test are normalized. In contrast, the presence of LA makes the results of the mix to remain prolonged. Finally, the presence of LA is likely quenched by increasing the PL concentration, thus providing confirmation of the diagnosis (Table 1).

As mentioned, the aPTT was the historical choice to screen patients for LA, but later such other PL-dependent tests as the dilute Russel viper venom test (dRVVT), dilute prothrombin time, silica clotting time (SCT) and others were introduced. The above tests possess variable sensitivity and specificity to detect LA and this variability is explained by two key issues: the multifaceted properties of LA and its interaction with methods/reagents that vary in terms of types and concentrations of PL (e.g., ethanolamine, phosphatidyl serine, etc.) and activators (e.g., kaolin, silica, ellagic acid, etc.). Although, general rules cannot be given for the choice of the most sensitive and specific methods/reagents to detect LA, I discuss in the next paragraph merits and drawbacks of the PL-dependent tests.

## 3. Recommended Tests to Detect LA

The recent guidelines for laboratory detection of LA, issued by ISTH recommend using at least two tests based on different principles: the aPTT (or aPTT-derived tests such as the SCT known to be sensitive to LA) and dRVVT [7]. The use of other tests and/or the use of multiple tests are discouraged, as this may lead to difficult interpretation of results and to the risk of incurring into false positive results that may induce clinicians to start (unnecessary) long term anticoagulation. Both aPTT-derived tests and dRVVT should be used under strict standardized conditions and their results should be expressed as ratio (patient-to normal clotting time) by testing a pooled normal plasma (PNP) in parallel with the patient plasma. The screen procedure is suggestive of LA whenever the results are prolonged beyond the cut off established in the laboratory by using the same method/reagent.

The mixing procedure should be performed on the screening test, which was abnormal and is indicative (not affirmative) of LA whenever the clotting time is closer to the patient rather than to the normal clotting time.

Finally, the confirm procedure should be performed on the screen test, which was abnormal by increasing the PL concentration and is affirmative for LA whenever the clotting time is closer to that of normal rather than to that of the patient.

Because of the multiform interaction between LA and PL-dependent tests, aPTT and dRVVT are concomitantly positive in a relatively small proportion of patients and therefore current recommendations mandate that LA is positive whenever one of the two tests is positive. Positivity for both tests could be taken as an evidence on the strength of LA [8]. There is emerging evidence that an additional test could be added to the panel of APS testing. Solid-phase antibodies to the complex phosphatidylserine (PS)–prothrombin (PT), known as aPS–PT could be a suitable addition to the existing tests and would identify patients with APS at high risk [9]. However, presently aPS–PT is not yet endorsed by APS guidelines.

## 4. Performance of LA Detection Procedures

Since the laboratory detection of LA is based on indirect evidence stemming from PL-dependent tests combined with laboratory criteria, it is paramount for clinicians and laboratory operators to be aware of the crucial issues that make the interpretation of results difficult.

### 4.1. Screen

This procedure is heavily affected by preanalytical and analytic variability. Plasma used for testing stems from blood anticoagulated at the proportion of 9:1 (blood:anticoagulant) with trisodium citrate, which may be commercially available at concentrations ranging from 0.105 to 0.129 M. Citrate concentration is highly effective in modifying the clotting time and therefore strict adherence to the same concentration is required for correct interpretation of results over time; current recommendations mandate citrate concentrations ranging from 0.105 to 0.109 M.

Residual platelets in the supernatant plasma are heavily dependent on speed and time of centrifugation and may have detrimental effect on test results. Since platelets (upon activation) express on their surface negatively charged PL, which are targeted by LA, excess residual platelets may quench LA activity. Consequently, low titer LA could be lost at diagnosis. This quenching effect is particularly relevant if test plasma is stored frozen for later use; freezing/thawing results in fact into platelet fragmentation and consequently to increased soluble PL concentrations. Current guidelines recommend double centrifugation: blood and supernatant plasma both at 2000× *g* for 15 min at (controlled) room temperature.

The choice of the type of aPTT and dRVVT is crucial for LA detection. There are many commercial brands available, and they vary in composition of PL and activators. This varied composition is inevitably reflected in the sensitivity and specificity of LA detection. As a rule, reagents with relatively low PL content are more sensitive to LA. Specificity is hard to define in the absence of specific test to detect “true” LA. However, it has been observed that LA detected by dRVVT is more likely than that detected by aPTT to be associated with clinical events [10]. However, this contention remains to be confirmed. Activators also play a crucial role for LA detection. Kaolin has been considered for many years as the most sensitive activator for the aPTT, but it is less frequently used because of interference with the optical clot detection system of modern coagulometers and sedimentation in the tubes connecting reagents reservoirs and cuvettes. More recently, silica has been used as a surrogate for kaolin as (although particulate) it does not interfere with coagulometers and still performs relatively well in terms of sensitivity. Finally, ellagic acid is a soluble activator that does not interfere with coagulometers, but its sensitivity to LA is still debated [11]. Current guidelines recommend silica as the activator of choice, although ellagic acid when combined with the appropriate PL concentrations may be used for LA detection [7].

Although no strict recommendations have been issued on the best composition and concentration of PL to detect LA, it is widely recognized that relatively low concentrations and synthetic PL are preferable [7].

Delay of testing after blood collection and centrifugation may be another crucial issue for the quality of results. Personal observations suggest that testing should be performed on fresh plasma (within three hours from collection); if this is not feasible, plasma should be (quickly) frozen and stored at −70 °C.

### 4.2. Mix

This procedure should be performed by mixing equal portions of patient and pooled normal plasma (PNP) without incubation. Crucial issue for standardization of the mix procedure is the PNP, which should be prepared by pooling equal portions of platelet-poor plasma from at least 30 donors (male and females). The average content of the individual coagulation factors in the PNP should be close to 100% and the material should be kept relatively free from platelets (i.e., double centrifugation). Homemade PNPs prepared and stored frozen at −70 °C for later use is recommended. Alternatively, freeze-dried PNPs of commercial origin can be used if they fulfil the above requirements.

### 4.3. Confirm

This procedure requires repetition of the screen test upon increasing PL concentrations. Quantity and quality of PL have not been firmly established. In the past, preparations of platelet lysate were used as source of PL for confirm procedure, but they were later abandoned because of difficult preparation, standardization, and storage. Current guidelines state that PL should hopefully be from synthetic origin and may have bilayer or hexagonal conformation [7].

## 5. Integrated Assays

There are commercially available assays for LA, which are based on dual tests (aPTT and dRVVT) carried out simultaneously at low (screen) and high (confirm) PL concentrations. Many laboratories are now adopting these integrated assays skipping the mix procedure. Results are interpreted directly from the screen/confirm ratio and positive LA is likely when the ratio is higher than cut off. The main advantage of integrated assays rests on the standardized preparation of screen and confirm components with minimal reagent handling. This simplification usually skips the mix procedure, assuming that LA and coagulation factor deficiencies behave differently in the integrated assay (i.e., LA give rise to screen/confirm ratios higher than cut off and coagulation factor deficiencies to ratios smaller than cut off). Although in most situations this procedure might be valid, there are certain types of LA that behave peculiarly. To express their anticoagulant activity, they require a plasma co-factor (called lupus co-factor [12]) that could be occasionally absent in the patient plasma. In this situation LA cannot prolong adequately the clotting time of the screen procedure but the prolongation becomes much more evident in the mix procedure when the normal plasma provides adequate amount of the missing co-factor. In those instances, skipping the mix procedure increases the chance of missing the diagnosis of this particular type of LA. The identity of the lupus co-factor has not yet been accurately established although there are hints that favor prothrombin (factor II) as the most probable candidate [12]. The incidence of this peculiar lupus co-factor in the general population of patients positive for LA is unknown and is probably rare. The author’s personal advice would be to adopt integrated assays for LA without skipping the mix procedure.

## 6. Results Expression and Interpretation

Results expression is an important step, especially for LA detection given the fact that the diagnosis is not straightforward. Clotting times stemming from screen and mix LA detection procedures should be normalized, dividing the patient clotting time by that of the PNP run in the same conditions and working session. The resultant screen-ratio and mix-ratio allow a uniform reporting over time and minimizes between-assay variability. Likewise, confirm procedures should be assessed as the ratio of screen-ratio-to- confirm-ratio. Results interpretation should be as follows.

### 6.1. Screen

Results are suggestive for LA (or coagulation factor deficiency) if the clotting time ratio for patient is higher than cut off.

### 6.2. Mix

Results are better interpreted taking into consideration the index of circulating anticoagulant (ICA), which can be calculated from the clotting time (CT) of mix, PNP and patient as follows:ICA = [(CT_mix_ − CT_PNP_)/CT_patient_] × 100(1)

Results are suggestive of circulating anticoagulant (either LA or a specific inhibitor to coagulation factor) if ICA is higher than cut off.

### 6.3. Confirm

Results are better interpreted taking into consideration the percent correction, which can be calculated as:%Correction = [(CT_screen_ − CT_confirm_)/CT_screen_] × 100(2)

Results are confirmatory for LA if the percent correction is higher than cut off.

## 7. Cut-Off Values

Results of LA testing must be compared with cut off values determined for screen, mix, and confirm to establish whether the patient is positive or negative. Cut off values may be determined as the value correspondent to the mean + 2 standard deviations, calculated from the results distribution of an adequate number of healthy subjects. However, clotting time results are rarely “normally” distributed and therefore cut off values should be better defined as the value corresponding to the 95th centile of the distribution. Non conclusive recommendations have been issued on the numbers of healthy subjects to be used to calculate cut off values [7]. Theoretically, 120 or more subjects should be enrolled, but this is unpractical, especially in small laboratories. Alternatively, cut off values determined by manufacturers of LA detection systems could be used if they are checked locally for accuracy using a smaller number of subjects (20 or 40). Recent observations indicate that cut off values determined in different laboratories enrolled their own 120 healthy subjects, who were tested by a wide representation of commercial LA platforms, differed even when the same platform was used in different laboratories [13]. These observations strongly argue against the use of cut off value determined elsewhere even when using the same platform.

## 8. Selection of Patients to Be Investigated

Selecting appropriate patients for LA detection is a crucial issue as the indiscriminate search may give rise to unacceptable numbers of false positive results. Current guidelines recommend performing laboratory investigation in patients with previous history of venous and/or arterial thrombosis, especially when occurring at young age and in the presence of autoimmune diseases. Pregnancy complications is another setting to be investigated for LA. Finally, LA should be searched in patients with (accidentally found) aPTT prolongation at the time of presurgical screen or other investigations. Detailed discussion on other conditions that require laboratory investigation for LA can be found elsewhere [7].

## 9. Timing of Testing

Timing of testing may be crucial as there may be situations when results interpretation is difficult. Current guidelines recommend avoiding LA investigation in patients during acute thrombotic events. Some of the coagulation factors (namely factor VIII) in addition to be strong procoagulants are also acute-phase reactants and are therefore increased during acute thrombosis; this may make results interpretation difficult. The same is valid for C-reactive protein (another acute-phase reactant), which may affect the aPTT results [14]. Notable exceptions to the above rule are patients presenting with acute arterial thrombosis. In this circumstance, clinicians may need prompt diagnosis of LA to make decision on the appropriate treatment to prevent recurrent events (either aspirin or oral anticoagulation). Finally, during acute events it is likely that patients referred to the laboratory have already been started on anticoagulation (heparins, vitamin K antagonists (VKA) or direct oral anticoagulants (DOAC)), which may make results interpretation difficult (see below). LA detection during pregnancy may also result in difficult interpretation as many coagulation factors are physiologically increased.

## 10. LA Detection in Anticoagulated Patients

Anticoagulants of any species (parenteral or oral) invariably prolong the clotting times of PL-dependent tests and therefore the interpretation of screen, mix, and confirm would be inherently difficult in patients who have already been started on anticoagulation. Until recently, LA detection was deemed not strictly needed during anticoagulation as VKA was considered the mainstay of treatment to prevent recurrent thrombotic events and usually LA detection was deferred until discontinuation of the regular course of VKA. However, more recently clinical trials of patients with APS randomized to receive rivaroxaban or VKA, showed that patients on rivaroxaban had unacceptable rates of recurrent thrombosis when compared to VKA [15,16]. One of these trials was prematurely interrupted [15] and this prompted the European Medicines Agency (EMA) to warn against using DOAC of whatever origin in patients with APS [17]. Some issues concerned with the use of DOAC in APS should however be considered. For example, the results of the clinical trials that prompted EMA decisions were obtained in patients treated with rivaroxaban and is still unknown whether the other DOAC have the same effect. Furthermore, patients enrolled in the above trials were triple positive for APS and therefore whether the same conclusions are valid for those patients with double or single APS positivity is still unknow. Nevertheless, following EMA indications, patients who experienced acute thrombosis after unspecified causes and have been started on rivaroxaban or other DOAC, should be promptly diagnosed for APS (including LA) and if positive, must be switched to VKA. Guidelines on how to diagnose LA in anticoagulated patients suggest a number of options discussed elsewhere [18], which are briefly summarized as follows (see also Table 2).

### 10.1. Unfractionated Heparin (UFH)

Patients on UFH may be diagnosed without treatment interruption if the laboratory system used for LA detection contains heparinase or polybrene, that are chemicals able to quench the effect of UFH up to 1 U/mL. Nonetheless, whenever the presence of UFH is suspected in the patient plasma, this can be ruled out by performing the thrombin clotting time, which is extremely sensitive to UFH, but not to LA.

### 10.2. Low Molecular Weight Heparin (LMWH)

Patients on LMWH may also be diagnosed during treatment as most aPTT reagents are insensitive to LMWH, although caution should be exerted when using brands of LMWH known to prolong the aPTT.

### 10.3. Vitamin K Antagonists (VKA)

Patients on VKA if tested by aPTT or dRVVT is not recommended as it may give rise to false negative or false positive LA, unless their plasma is mixed with equal amounts of normal plasma prior to testing. The mixture should be able in most instances (especially if the INR is <3.0) to correct the VKA-induced coagulation defect, thus allowing for LA detection without significant VKA interference. It should however be considered that coagulation defects induced by VKA might not be completely corrected upon mixing. Furthermore, mixing (by definition) reduces the potency of LA by 50%. Hence, weak LA may be lost at diagnoses. All in all, diagnosis of LA in patients on VKA if performed should be interpreted with caution.

### 10.4. Direct Oral Anticoagulants (DOAC)

DOAC create unprecedented issues for LA diagnosis as coagulation defects induced by these drugs cannot be corrected neither by heparinase/polybrene nor by mixing patient and normal plasma. Recent observations from the literature showed that patients on DOAC may erroneously be classified as LA positive, especially when the drug is rivaroxaban, and LA detection is performed with dRVVT [19,20]. A possible solution is removing DOAC by active charcoal chemicals (absorbents) that proved effective in removing DOAC from plasma. These absorbents are mixed with plasma; after short incubation, the mixture is centrifuged, and LA detection performed on supernatant plasma. Experimental data show that commercially available DOAC absorbents can remove any kind of DOAC from plasma and perform relatively well for LA detection in most instances [19,20]. These absorbents represent a significant step forward to diagnosing LA in patients on DOAC. However, it should be realized that small amounts of DOAC may remain in the supernatant after exposure to the absorbents which may affect results. Moreover, further investigation in patients truly positive for LA prior initiating anticoagulation is required. If none of the above options is feasible, it is pragmatically suggested to test anticoagulated patients for aCL and aβ2-GP-I, which being solid-phase antibodies are not influenced by anticoagulation. If both or one of these antibodies are positive for the IgG isotype, the patients could be pragmatically considered APS positive and the patient if anticoagulated with DOAC should be switched to VKA.

## 11. Distinguishing LA from Inhibitors to Coagulation Factors

Distinguishing LA from inhibitors to coagulation factors is not always possible based on the interpretation of results stemming from PL-dependent tests and diagnostic procedures (screen, mix and confirm). Typical example is the inhibitor to factor VIII, frequently found in patients with autoimmune diseases (acquired hemophilia). These inhibitors are thought to exert their inhibitory activity upon incubation (2 h at 37 °C) of the mixture with the PNP and therefore they could be undetected when testing the mixture without incubation (as it is usually done for LA detection). However, when the inhibitors to factor VIII possess high potency (as they occur in patients with acquired hemophilia), are likely to inhibit immediately the factor VIII provided by the PNP used in the mix procedure and therefore, it is not possible to distinguish them from LA. To make things worse, some inhibitors to factor VIII behave like LA in the confirm procedure (i.e., the clotting time upon increasing PL concentrations is corrected) [21]. The best way to distinguish LA from specific inhibitors to coagulation factors is knowledge of the clinical history of the patient being investigated: patients with inhibitors to coagulation factors usually bleed; patients with LA (with few exceptions) experience thrombotic events.

## 12. Result Reporting

Result reporting is an important step given the fact that the LA diagnosis is not straightforward. Current guidelines recommend that laboratories report analytical results for the three diagnostic procedures (screen, mix and confirm) along with cut off values and a conclusive statement on whether results are compatible or not with positivity for LA. This conclusive statement is paramount, as clinicians may not be aware of the complex diagnostic procedures, which are behind LA detection. According to guidelines, clinicians should also be informed that the laboratory diagnosis (if positive) must be confirmed 12 weeks apart as LA positivity in some patients may be transient and is not of clinical significance [7] Finally, results should be interpreted, validated, and reported in combination with the other APS parameters (i.e., aCL and a β2-GP-I), keeping in mind that triple positivity identify patients at high risk.

## 13. Concluding Remarks

Testing for LA is a challenging task for the clinical laboratory because specific tests for its detection are not yet available. However, LA detection is paramount for patients’ management, as its persistent positivity in the presence of (previous or current) thrombotic events, candidate patients for long term anticoagulation. Guidelines for LA detection have been established and updated over the last two decades [7]. Implementation of these guidelines across laboratories and participation to external quality assessment schemes are urgently required to help standardize the diagnostic procedures and help clinicians for appropriate management of the relatively rare but potentially devastating APS condition.

## Figures and Tables

**Figure 1 biomedicines-09-00844-f001:**
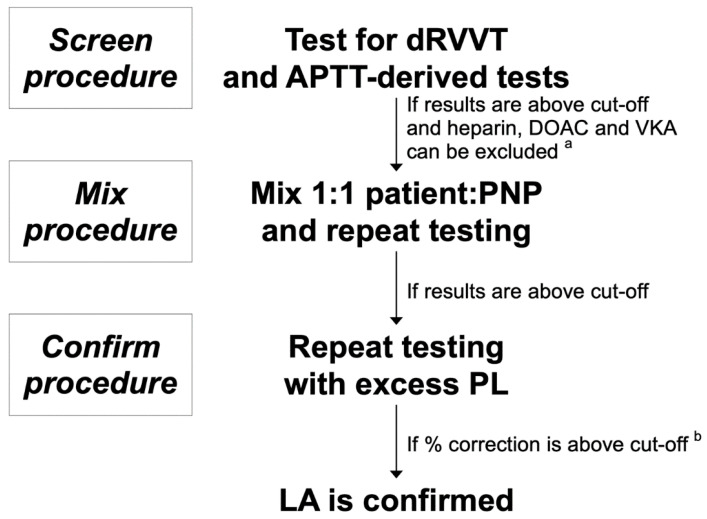
Three-step diagnostic procedure for LA. ^a^ Prolonged thrombin clotting time suggests the presence of unfractionated heparin; the presence of LMWH or DOAC should be confirmed by the anti-FXa or DOAC assays, respectively. ^b^ Confirmation can be performed on a mixture of patient and normal plasma if the confirm clotting time on undiluted plasma is prolonged. LA, lupus anticoagulant. dRVVT, dilute Russel viper venom test. aPTT, activated partial thromboplastin time. PNP, pooled normal plasma. PL, phospholipids. LMWH, low molecular weight heparin. DOAC, direct oral anticoagulants. VKA, vitamin K antagonists.

**Table 1 biomedicines-09-00844-t001:** Diagnostic criteria to detect lupus anticoagulant (LA).

Diagnostic Criteria	Rational	Outcome
Screen	LA targets and inhibits negatively charged phospholipids of aPTT/dRVVT	Prolongation of the clotting time
Mix	Pooled normal plasma provides excess coagulation factors to the test plasma	Clotting time remains prolonged in the presence of LA and is shortened in the presence of coagulation factors deficiency
Confirm	Excess phospholipids quench LA	Clotting time is shortened in the presence of LA

**Table 2 biomedicines-09-00844-t002:** Main options to detect lupus anticoagulant (LA) in anticoagulated patients.

Patient Anticoagulated with:	Diagnostic Option
Unfractionated heparin (UFH)	LA testing can be performed whilst patients are on treatment if the LA detection test contains heparinase or polybrene to neutralize UFH up to 1U/mL
Low molecular weight heparin (LMWH)	LA testing can be performed whilst patients are on treatment as aPTT is relatively insensitive to LMWH. Caution should however be exerted in results interpretation.
Vitamin K antagonists (VKA)	LA testing on undiluted plasma is not recommended whilst patients are on treatment. Testing can be performed upon dilution (1:1) of patient plasma into pooled normal plasma (PNP) as PNP is able to correct the coagulation factors deficiency induced by VKA (if the INR is <3.0). Caution should be exerted because of possible false positive or false negative results
Direct oral anticoagulants (DOAC)	LA testing on undiluted plasma is not recommended. Testing can be done upon removal of DOAC by absorbents (see text for more details)

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
