# Peer review of "Diagnostic Challenges on the Laboratory Detection of Lupus Anticoagulant"

_biomedicines, 2021, doi:10.3390/biomedicines9070844_

Round 1

Reviewer 1 Report

This is a comprehensive and valuable review on the laboratory challenges for detecting and characterizing Lupus Anticoagulant (LA), and beyond Anti-Phospholipid-Syndrome (APS). This report is well-structured, well-presented and documented, and is pleasant to read. It addresses key practical issues that laboratories facing this diagnosis have to correctly control.

Concerning global comments, I just would like to keep the attention that clotting tests for LA are extensively and accurately presented and discussed by the author, and laboratory issues extensively described. The author gives useful recommendations for a good laboratory practice. However, the anti-phospholipid side is only evoked, and is described too rapidly. As clotting methods and immunoassays for anti-phospholipid-antibodies (APA) are frequently used together for investigating patients with APS or suspected, a more extended description of those immunoassays, and a discussion on their practice and limitations would be useful. The paper is focused on LA, but diagnosis of LA using clotting methods usually involves immunoassays. In addition, the guidelines recommend to use the 2 clotting systems, APTT and dRVVT as described by the author, and 2 immunoiassays, anti-cardiolipin plus anti-ß2GP1. Presenting the complete diagnostic approach should improve the message of this article.

In addition, when testing LA or APAs in LA/APS suspected patients, the timing is important (chapter 9). However, this timing does not concern only the patient's environment and therapy, which could impact LA testing, but diagnosis requires to confirm the test results at least twice, 12 weeks apart (to avoid transitory anticoagualants or antibodies). This would be useful to remember this requirement in this article.

For specific comments, I just would like to suggest mentionning the normalized LA ratio at line 110, as it is alreday indicated in line 211.

Author Response

Dear Editors,

I am resubmitting the manuscript that has been revised according to the comments made by the reviewers. Those parts that have been changed are underlined in the text. Here my point-to-point responses to the comments made by the reviewers.

Reviewer 1

This is a comprehensive and valuable review on the laboratory challenges for detecting and characterizing Lupus Anticoagulant (LA), and beyond Anti-Phospholipid-Syndrome (APS). This report is well-structured, well-presented and documented, and is pleasant to read. It addresses key practical issues that laboratories facing this diagnosis have to correctly control.

Comment 1. Concerning global comments, I just would like to keep the attention that clotting tests for LA are extensively and accurately presented and discussed by the author, and laboratory issues extensively described. The author gives useful recommendations for a good laboratory practice. However, the anti-phospholipid side is only evoked, and is described too rapidly. As clotting methods and immunoassays for anti-phospholipid-antibodies (APA) are frequently used together for investigating patients with APS or suspected, a more extended description of those immunoassays, and a discussion on their practice and limitations would be useful. The paper is focused on LA, but diagnosis of LA using clotting methods usually involves immunoassays. In addition, the guidelines recommend to use the 2 clotting systems, APTT and dRVVT as described by the author, and 2 immunoassays, anti-cardiolipin plus anti-ß2GP1. Presenting the complete diagnostic approach should improve the message of this article.

Answer. Discussion on immunoassays is not the scope of this review which is entirely devoted (see title of the manuscript) to LA. I understand the reviewer concern and add a few words and a new reference (see page 2).

Comment 2. In addition, when testing LA or APAs in LA/APS suspected patients, the timing is important (chapter 9). However, this timing does not concern only the patient's environment and therapy, which could impact LA testing, but diagnosis requires to confirm the test results at least twice, 12 weeks apart (to avoid transitory anticoagulants or antibodies). This would be useful to remember this requirement in this article.

Answer. Amended (see page 1).

Comment 3. For specific comments, I just would like to suggest mentioning the normalized LA ratio at line 110, as it is already indicated in line 211.

Answer. The line to which the reviewer is alluding pertains to a general statement on clotting time and does not require “the ratio”.

Reviewer 2 Report

This review interestingly summarizes what is known about LA diagnosis and what remains to be optimized. LA diagnosis is quite challenging for laboratory operators and clinicians due to many crucial issues that interfere with the results and the direct impact of the latter on patient management. The author develops and discusses the different available tests and diagnostic strategy, the performance of each assay, the pre-, post- and analytical conditions and the results expression as well as the main interferences that laboratory operators should be aware of along with the areas that remain to be optimized. Some concerns warrant further attention by the author.

1/- Page 1, L 37-40: In the definition of the APS, two important issues are worth to be specified: 1/- the min and max time-interval between the clinical and the biological criteria of APS and 2/- the persistent positivity (≥ 12 weeks’ interval) of antiphospholipid antibodies.

2/- Figure 1: why only heparin should be excluded if the screen results are above the cut-off and not also DOAC and VKA therapies?

3/- Page 3, L 84: Most probably Table 1 should be referenced and not Table 2.

4/- Page 3, L100: according to the ISTH guidelines, any aPTT could be used for LA diagnosis or LA sensitive aPTT should be specifically employed? This deserves to be specified.

5/- Page 4, L153-154: The author mentioned that Kaolin is no longer used. This notion needs to be seen since reagents with kaolin as activator are still widely used, even with modern coagulometers, particularly those based on mechanical clot detection. Its interference with the optical clot detection system deserves more detailed explanation.

6/- Are freeze-dried, frozen “home-made” and frozen commercialized PNPs equivalent? In other words, are laboratory operators free to choose any of these 3 options with no preference of one over the others?

7/- Page 5, L190: “cu off” should be corrected.

8/- Paragraph 6: what about the normalized ratio, which consists on calculating the ratio of the screen ratio/confirm ratio? Why this is not mentioned?

9/- Page 6, L240: concerning the “smaller number of subjects”, this deserves more specifications. Is there any minimum number to be used?

10/- Table 2: “ca be” should be corrected in UFH and LMWH lines. Apart from that, in UFH line, to “… if LA detection test contains heparinase or polybrene to neutralize UFH” should be added that this is true up to 1 U/mL.

11/- Table 2 & paragraph 10.2: for LMWH, relatively insensitive aPTT could be used independently of LMWH concentration? What about dRVVT? Should heparinase be added to these reagents for LA testing in LMWH samples?

12/- Table 2 & paragraph 10.3: the mixing study could be considered in VKA samples if INR value is within 1.50 and 3.0 interval otherwise it shouldn’t. This deserves to be mentioned.

13/- Paragraph 10.4: DOAC adsorbents can completely remove DOAC compounds. However, many studies showed that this may also not be the case, especially that low DOAC concentrations may still be present and interfere with LA results like for instance in Jourdi et al. Thromb Res 2019 and Farkh et al. Front Med 2021. This deserves to be mentioned especially that false positive LA results cannot be ruled out in this case even after using DOAC adsorbent devices.

Author Response

Reviewer 2

This review interestingly summarizes what is known about LA diagnosis and what remains to be optimized. LA diagnosis is quite challenging for laboratory operators and clinicians due to many crucial issues that interfere with the results and the direct impact of the latter on patient management. The author develops and discusses the different available tests and diagnostic strategy, the performance of each assay, the pre-, post- and analytical conditions and the results expression as well as the main interferences that laboratory operators should be aware of along with the areas that remain to be optimized. Some concerns warrant further attention by the author.

Comment 1.- Page 1, L 37-40: In the definition of the APS, two important issues are worth to be specified: 1/- the min and max time-interval between the clinical and the biological criteria of APS and 2/- the persistent positivity (≥ 12 weeks’ interval) of antiphospholipid antibodies.

Answer. Amended (see page 1).

Comment 2/- Figure 1: why only heparin should be excluded if the screen results are above the cut-off and not also DOAC and VKA therapies?

Answer. Amended (see Fig).

Comment 3/- Page 3, L 84: Most probably Table 1 should be referenced and not Table 2.

Answer. Amended.

Comment 4/- Page 3, L100: according to the ISTH guidelines, any aPTT could be used for LA diagnosis or LA sensitive aPTT should be specifically employed? This deserves to be specified.

Answer. Amended (see page 3).

Comment 5/- Page 4, L153-154: The author mentioned that Kaolin is no longer used. This notion needs to be seen since reagents with kaolin as activator are still widely used, even with modern coagulometers, particularly those based on mechanical clot detection. Its interference with the optical clot detection system deserves more detailed explanation.

Answer. Besides optical interference, kaolin sediments along the tubes connecting cuvette and reagents reservoir. This has been added (see page 5).

Comment 6/- Are freeze-dried, frozen “home-made” and frozen commercialized PNPs equivalent? In other words, are laboratory operators free to choose any of these 3 options with no preference of one over the others?

Answer. Amended (see page 5).

Comment 7/- Page 5, L190: “cu off” should be corrected.

Answer. Amended.

Comment 8/- Paragraph 6: what about the normalized ratio, which consists on calculating the ratio of the screen ratio/confirm ratio? Why this is not mentioned?

Answer. Amended (see page 6).

Comment 9/- Page 6, L240: concerning the “smaller number of subjects”, this deserves more specifications. Is there any minimum number to be used?

Answer. Specified (see page 6).

Comment 10/- Table 2: “ca be” should be corrected in UFH and LMWH lines. Apart from that, in UFH line, to “… if LA detection test contains heparinase or polybrene to neutralize UFH” should be added that this is true up to 1 U/mL.

Answer. Amended.

Comment 11/- Table 2 & paragraph 10.2: for LMWH, relatively insensitive aPTT could be used independently of LMWH concentration? What about dRVVT? Should heparinase be added to these reagents for LA testing in LMWH samples?

Answer. Relatively insensitive aPTT to LMWH are OK. Most reagents for dRVVT do contain polybrene or heparinase.

Comment 12/- Table 2 & paragraph 10.3: the mixing study could be considered in VKA samples if INR value is within 1.50 and 3.0 interval otherwise it shouldn’t. This deserves to be mentioned.

Answer. Mentioned (see page 8 and Table 2).

Comment 13/- Paragraph 10.4: DOAC adsorbents can completely remove DOAC compounds. However, many studies showed that this may also not be the case, especially that low DOAC concentrations may still be present and interfere with LA results like for instance in Jourdi et al. Thromb Res 2019 and Farkh et al. Front Med 2021. This deserves to be mentioned especially that false positive LA results cannot be ruled out in this case even after using DOAC adsorbent devices.

Answer. This part has been rephrased accordingly (see page 9).

I thank the reviewers for helpful comments and look forward to hearing from you.

Best regards

A Tripodi